# Potential Nanotechnology-Based Therapeutics to Prevent Cancer Progression through TME Cell-Driven Populations

**DOI:** 10.3390/pharmaceutics15010112

**Published:** 2022-12-29

**Authors:** Rafia Ali, Huimin Shao, Pegah Varamini

**Affiliations:** 1School of Pharmacy, Faculty of Medicine and Health, University of Sydney, Sydney, NSW 2006, Australia; 2The University of Sydney Nano Institute, The University of Sydney, Sydney, NSW 2006, Australia

**Keywords:** tumour microenvironment, nanoparticles, triple negative breast cancer, targeting, tumour progression, metastasis, therapeutic resistance

## Abstract

Triple-negative breast cancer (TNBC) is the most aggressive subtype of breast cancer with a high risk of metastasis and therapeutic resistance. These issues are closely linked to the tumour microenvironment (TME) surrounding the tumour tissue. The association between residing TME components with tumour progression, survival, and metastasis has been well elucidated. Focusing on cancer cells alone is no longer considered a viable approach to therapy; thus, there is a high demand for TME targeting. The benefit of using nanoparticles is their preferential tumour accumulation and their ability to target TME components. Several nano-based platforms have been investigated to mitigate microenvironment-induced angiogenesis, therapeutic resistance, and tumour progression. These have been achieved by targeting mesenchymal originating cells (e.g., cancer-associated fibroblasts, adipocytes, and stem cells), haematological cells (e.g., tumour-associated macrophages, dendritic cells, and myeloid-derived suppressor cells), and the extracellular matrix within the TME that displays functional and architectural support. This review highlights the importance of nanotechnology-based therapeutics as a promising approach to target the TME and improve treatment outcomes for TNBC patients, which can lead to enhanced survival and quality of life. The role of different nanotherapeutics has been explored in the established TME cell-driven populations.

## 1. Introduction

Triple-negative breast cancer (TNBC) is a highly aggressive subtype, constituting 15–20% of all breast cancers and having the highest risk of relapse, metastasis, and mortality rate [1]. The immunohistochemical characteristics reveal a lack of oestrogen receptors, progesterone receptors, and the human epidermal growth factor 2 (HER2) receptor. This lack of receptors limits the therapeutic options available for TNBC patients leaving them with limited targeted therapies [2]. The current main modalities of treatment include general chemotherapy and radiotherapy [3]. However, chemotherapy is associated with significant side effects, due to its off-target effects, in conjunction with the inherent and acquired therapeutic resistance that pose as substantial barriers to treatment [4].

The aggressive nature of TNBC is closely related to the tumour microenvironment (TME), impacting tumour initiation, survival, and propagation. The recent shift in oncology studies has provided the foundation for understanding the dynamic nature of the TME and highlighted the limitations of targeting tumours alone. The heterogeneous stroma harbours various cellular and non-cellular components that provide structural and functional support [5]. These can be categorised into three major components: (1) mesenchymal originating cells, (2) haematological cells, and (3) non-cellular factors (Figure 1).

Components related to mesenchymal cells include cancer-associated fibroblasts, tumour-associated adipocytes, and stem cells. Immunological cells pertain to tumour-associated macrophages, dendritic cells, neutrophils, CD8+T cells, and CD4+T cells, and mast cells. Lastly, the favourable tumour environment is further established through non-cellular structures such as the extracellular matrix [6]. Moreover, the release of various cytokines, interleukins (IL), chemokines, and growth factors all aid in tumoral behaviour and oncogenesis [7]. Despite the complicated interplay of the TME, a benefit is that it is genetically stable compared with breast cancer cells [8]. Therefore, selective targeting of the TME is imperative in improving TNBC patient outcomes. 

Various elements of the tumour surroundings have adapted to facilitate the needs of the heterogeneous cancer cells that further lead to tumour survival and progression. For instance, tumour cells that are away from the vessels lack oxygen and essential nutrients which are required for uncontrolled proliferation. To mitigate this issue, tumours and the TME cells secrete factors such as vascular endothelial growth factor (VEGF) which promote angiogenesis [9]. However, the architecture of these vessels significantly differs from normal vascular structures due to the apparent gaps between the endothelial cells [9]. The hyper-permeability results in the perfusion of both immune and non-immune related cells into the TME, with the benefit of allowing nanoparticle entry [10]. Moreover, tumour cells take advantage of reactive oxidative species, generally involved in oxidizing and damaging proteins to aid in proliferation [9]. Several redox mechanisms are involved in multiple drug resistance (MDR). It is suggested that redox-active drugs (antioxidants and prooxidants) can be used to overcome MDR [11]. 

Nanotechnology-based drug delivery systems are among the emerging approaches that are intensively explored in the treatment of cancer [12]. The application of nanoparticles (NPs) is promising in TNBC due to their favourable physiochemical characteristics and preferential tumour accumulation. Some of the different types of nanoparticles include micelles, liposomes, extracellular vesicles, and gold nanoparticles [13]. These NPs can encapsulate drugs that may have limited application due to their hydrophobicity or unstable nature. NPs’ surfaces can be engineered with different moieties that allow for active targeting (Figure 2) [14]. Despite biological and functional barriers created by the TME, NPs have been shown to improve the intra-tumoural accumulation of drugs through passive means or active targeting. Passive targeting refers to the enhanced penetration and retention effect (EPR) that relies on the leaky vasculature seen within the TME. The reduced biodistribution to the non-cancerous tissues leads to fewer side effects, adding to their benefit in optimising cancer therapy [14]. 

Some NPs have shown to deliver consistent quantities of medications at controlled rates to the tumour sites [8]. Additionally, these nano-based carriers can encapsulate hydrophobic drugs that may have limited therapeutic use otherwise [7]. Studies have found that for optimal penetration through to the TME and enhanced tumour access, the nanoparticles size should be <50 nm [7]. Targeting different elements within the TME using NPs poses a desirable approach to ameliorating therapeutic resistance. In this review, various components of the TME and, subsequently, the application of different nanotechnology-based platforms for the delivery of therapeutic agents to the TME will be discussed for the treatment of TNBC.

Due to the complexity of the TME, the characteristics of different contributing cells are still under investigation, with some having controversial roles. For instance, while mast cells are present within the TME, their primary role is in allergy by releasing histamine, but their effects in cancer are yet to be elucidated. Recent studies have highlighted that mast cells contribute to tumour growth and progression [15]. There is conflicting evidence in that some suggest mast cells provide positive outcomes, whilst others argue it enhances tumour progression through the release of VEGF [15]. Annexin 1 (ANXA1) is a protein that has been identified to promote distant metastasis and regulate mast cell reactivity. Studies have found that TNBC patients with ANXA1-positive tumour cells had significantly worse overall survival compared with those who were ANXA1-negative [15]. The study paved the way for future research to explore the association between mast cells and TNBC. Despite the presence of various records on the development of NP-based carriers to target different components of the TME, there is a gap in the literature regarding nanotechnologies that deliver therapeutic agents to mast cells for TNBC. The controversy might be a contributor to the lack of mast cell-targeted therapies using NPs. 

## 2. Mesenchymal-Originating Cells

### 2.1. Cancer-Associated Fibroblasts

Cancer-associated fibroblasts (CAFs) are one of the most abundant types of stromal cells. They have various roles, including the production of different factors and cross-communication with other cells within the TME. Generally, fibroblasts possess antitumour activity, but when recruited by the tumour cells, the secretion of hepatocyte growth factor influences the differentiation to cancer-associated fibroblasts [5,16]. CAFs can also be formed from either epithelial cells, bone marrow-derived stem cells that have undergone epithelial-mesenchymal transition (EMT), or due to breast tissue differentiation [5]. 

CAFs’ distinctive characteristic of expressing surface proteins such as fibroblast-associated proteins (FAP) and alpha-smooth muscle actin (a-SMA) make them potential targets for cancer therapy [17]. Primarily, CAFs facilitate tumour macrophage infiltration, collagen production, interferon activation, along with the production of multiple cytokines and growth factors such as VEGF, fibroblast growth factor 2 (FGF2), transforming growth factor β (TGFβ), C-X-C chemokine ligand 12, and interleukin 6 (IL-6) [18]. The endpoint results are tumorigenesis, metastasis, angiogenesis, therapeutic resistance, and ultimately poorer patient outcomes. This vast functionality of CAFs makes them a promising target for developing novel nanotherapeutics, hence, they have been extensively studied in different nano-based drug delivery systems. The investigation of potential applications is hypothesised to not only alleviate therapeutic resistance but also to improve antitumour responses [19]. 

Losartan is an angiotensin II receptor blocker that has shown to have antifibrotic effects. A study conducted by Musettia and Huang explored this hypothesis with losartan-loaded hydrogels that found a reduction in collagen and CAF populations [10]. The microenvironment remodelling was not indicative of tumour suppression, rather it enabled tumour vulnerability that resulted in better efficacy of chemotherapeutic agents and prolonged survival in an orthotopic model [10,20]. Additionally, another study demonstrated immunogenic death by using CAF-targeted peptide NPs in TNBC cells [10]. Applications of different NP platforms in CAF targeting has been discussed in further detail in the following section.

#### 2.1.1. H-Ferritin-Based NPs

The distinctive FAP expression solely on CAFs provides a positive medium for selectivity. Sitia et al. exploited the overexpression of FAP on CAFs through an H-ferritin-based NP system. These NPs were loaded with navitoclax (Nav), a pro-apoptotic small molecule that inhibits the pro-survival protein, BCL-2 [21]. Nav’s mechanism of action involves triggering effector proteins that permeabilise mitochondrial membranes, resulting in cell apoptosis [21]. However, the hydrophobicity of Nav and thrombocytopenia side effects are the primary reasons for its limited therapeutic application. Loading Nav onto NPs was proposed to mitigate these obstacles [22,23,24]

The surface of H-ferritin nanocages was functionalised with the fragment antigen-binding (Fab) region of an anti-FAP antibody (HNav-FAP). These were compared with non-functionalised NPs and tested on FAP-overexpressed CAF cells to determine penetration and antitumour properties. A significantly higher binding of HNav-FAP nanoparticles to CAFs was observed compared with bare NPs. There was also a quick internalisation of the NPs in FAP-positive CAFs with a high uptake of the payload. However, there was no cell viability reduction with FAP-negative CAFs, highlighting the specificity of surface manipulation in delivering the cytotoxic drug. The biodistribution was evaluated using 4T1 tumour-bearing mouse models. Systemic distribution was limited, and more predominant accumulation was observed at the tumour sites in HNav-FAP compared with the naked NPs. HNav-FAP also had a significant tumour reduction over the experimental period at 2 µM concentrations of Nav, with only 19% viable cells compared with non-functionalised NPs (32%) and the drug alone (35%) (*p* = 0.000115) [21].

#### 2.1.2. Gold Nanoparticles

A study conducted by Bromma et al. investigated the use of gold nanoparticles (GNP) and their potential effects on tumour cells, CAFs, and normal fibroblasts (Figure 3). Ultimately, the aim was to see antitumour activity against neoplastic cells and eradication of CAFs whilst not impacting normal fibroblasts. NPs (15 nm) were functionalised with a combination of polyethylene glycol (PEG) and Arg-Gly-Asp (RGD) peptide (the minimal integrin-binding motif and a cell adhesion peptide). The GNP complex’s (GNP_PEG-RGD_) rationale was to stabilise the NPs, prevent aggregation, and improve NP uptake by cells. It was found that while the addition of PEG reduced cell penetration, functionalising with RGD improved uptake significantly [18].

The presence of GNP in normal fibroblasts was found to be very limited and only seen after 24 h of incubation. GNP_PEG-RGD_ uptake within CAFs and TNBC cells was 6- and 12-fold greater than fibroblasts, respectively. The GNPs accumulated within the cytoplasm and not in the nucleus, transported by the microtubules of the cells. Furthermore, the retention of NPs was significantly greater in CAFs and TNBC cells than in fibroblasts. These findings suggested that the designed delivery system could be a promising selective approach for treating TNBC by not only targeting tumour cells but also the CAFs.

### 2.2. Mesenchymal Stem Cells

Among different heterogeneous populations of cells in the TME, cancer stem cells (CSCs) play an important role in cancer initiation and progression. Mesenchymal stem cells (MSC) are a specific population of cells that greatly contribute to the highly invasive nature of TNBC and its established metastatic properties [25]. MSCs have the characteristic of non-malignant stem cells with the additional capacity to recapitulate, renew the TME, and differentiate into other tumorigenic cell types. Targeting only tumour cells is not sufficient in most cases, particularly in TNBC, as stem cells can renew and cause relapse. Hence, the need for targeting MSCs is crucial. CD44/CD24 are among the CSC markers in breast cancer. Previous studies have shown that CD44/CD24 phenotype, basal-like status, and number of involved lymph nodes are among the most important factors in TNBC patients’ survival. Among all different MSC phenotypes, CD44+/CD24− patients have the poorest prognosis [26]. CD44 receptor overexpression is associated with invasive tumour behaviour and is indicative of cell motility, proliferation, and angiogenesis [27].

#### 2.2.1. Salinomycin/Doxorubicin-Loaded Liposomes

CSCs are sensitive to salinomycin antibiotics produced by Streptomyces bacteria. Studies have shown a reduction in CSCs after treatment with this antibiotic [28]; however, it has limited clinical use due to its hydrophobicity. Kim et al. conducted a study that aimed to ameliorate clinical outcomes of breast cancer patients by concurrently targeting CSCs and tumour cells using liposomes (Figure 4) [29]. In this study, cross-linked multilamellar liposomal vesicles (cMLV) were co-loaded with salinomycin and doxorubicin (cMLV (Dox + Sal)) and coated with PEG. CSCs were co-cultured with MDA-MB-231 (a TNBC cell line) and were treated with the NPs. A synergistic effect was found when the combination of the two drugs was used achieving the greatest reduction in tumour cells compared with doxorubicin alone and the antibiotic alone. An additional significant finding was an increase in the CSCs’ population after treatment with doxorubicin-loaded NPs together with a reduction in tumour cell viability. Conversely, salinomycin NPs did not have an impact on the tumour cell viability, rather it reduced the number of CSCs [29].

#### 2.2.2. Targeting CD44+ Receptors

The overexpression of CD44 receptors on CSCs can be exploited for selective targeting. The receptor expression was utilised by a study where hyaluronic acid (HA), a CD44 ligand, was conjugated onto NPs. The NPs were then encapsulated with curcumin and paclitaxel to exert antitumour activity. In MDA-MB-231 cell lines overexpressing CD44, the targeted nano-formulation greatly reduced CSCs populations [30]. 

In another study using HA for CD44 targeting, a co-delivery system of doxorubicin and cyclopamine was loaded into hyaluronic acid-cystamine-poly lactic-co-glycolic acid (PLGA) NPs. In TNBC cell lines, the dual targeting resulted in remarkable cytotoxicity to both tumour cells and CSCs [31].

### 2.3. Tumour-Associated Adipocytes

A distinctive characteristic of TNBC is the high abundance of adipocytes within the TME [32]. The phenotype of tumour-associated adipocytes (TAAs) differs from normal adipocytes due to their reduced size and overexpression of collagen [6]. Understanding the extent of adipocyte’s role in TNBC is poorly elucidated. However, it has been well established that adipocytes increase tumour survival and proliferation through metabolic processes, with the potential of being a tumour initiator [6,33]. One of the known mechanisms of proliferation is through the release of a chemokine, CC-chemokine ligand 5 (CCL5) [5]. It is known that providing motility to tumour cells will allow for their migration and dissemination to a great extent [5]. Studies have demonstrated that glucose may change the promoting ability of human adipocytes on TNBC cells’ invasiveness. In one study, adipocytes cultured in high glucose content had a two-fold increase in TNBC tumour activation and enhanced motility when compared with those cultured in a low-glucose environment, which represents normal fasting glucose levels in humans [32]. 

Moreover, the fatty acids and glycerol produced from lipids are utilised by tumours through fatty acid receptor CD36 for the biosynthesis of cellular membranes required for proliferation [34]. 

TAAs release CC-chemokine ligand 2 (CCL2), through which they cause immunosuppression and further recruit factors into the tumour microenvironment to facilitate tumour survival and propagation. The secretion of CCL2 recruits monocytes into the TME, which then undergo malignant transformations to myeloid-derived suppressor cells (MDSC) and macrophage 2 subtypes (M2). Both MDSC and M2 cells are potent immunosuppressors that exert their effects by inhibiting T cell infiltration and through the release of other mediators [33].

#### CCL2 Trap Encapsulated NPs

A study conducted by Liu et al. designed a nano-based delivery system with the rationale of reducing the over-secretion of CCL2, which corresponds to increased MDSC and M2 population and subsequently enhanced immunosuppression. They designed a protein trap that was bound with CCL2 [33]. Lipid-protamine-pDNA (LPD) NPs were developed to deliver plasmid DNA encoding the CCL2 trap (pCCL2) to the TME using aminoethyl anisamide amine ligand. Compared with the control, significant TME remodelling was identified in the pCCL2 trap NPs through a three-fold reduction in a-SMA expression, a four-fold decrease in collagen, and an improved CD3+ T cell infiltration. A fundamental outcome was the high binding affinity of the NPs to CCL2 which caused a reduction in tumour volume over the experimental period, with no evident toxicity to the healthy organs. This study also reported a significant reduction in MDSC and M2 populations, corresponding to a decreased CCL2 concentration. Previous studies have reported using monoclonal antibodies to reduce CCL2 concentrations. However, the studies were unsuccessful, as a rebound effect was seen after ceasing treatment with a higher subsequent acceleration of tumour progression [33]. The pCCL2-loaded LPD NPs were not associated with this issue, adding to their positive profile, and making this system a good candidate for delivering therapeutics to TAAs.

## 3. Haematological Cells

### 3.1. Tumour-Associated Macrophages

Tumour-associated macrophages (TAMs) are one of the most predominant innate immune cells residing within the TME. TAMs heavily contribute to immunosuppression, allowing tumour progression. Macrophages can be characterised into two subtypes: M1, which has immunostimulatory characteristics usually involved in inflammation and fighting against bacteria, and M2 subtypes which are immunosuppressive, and act in wound healing and tumour cell protection [33]. However, M1 is only typically seen during the early stages of cancer; as the disease progresses, the M2 subtype overrides the TME and becomes the more predominant phenotype [6]. Differentiation is closely linked to the tumour cells and the cytokines that the macrophages are exposed to. Mature M2 formation is driven by the colony-stimulating factor (CSF1) that is secreted by tumour cells. TAMs play a major role in angiogenesis, metastasis, and tumorigenesis [35]. Clinical observations report that increases in TAMs correspond to poorer prognosis [6]. TAMs also have downstream effects on other residing cells; for example, the STAT3 activation causes the differentiation of myeloid-derived suppressor cells into their pro-tumorigenic phenotype, which is involved in tumour initiation [36]. Early elimination of TAMs is hypothesised to reduce immunosuppression and preclude the progression of TNBC.

Changing the polarisation from M2 to M1 subtypes or directly inhibiting M2 has been a target of numerous NPs in previous research. For instance, Rodell et al. loaded toll-like receptor 7/8 agonists into NPs to initiate the polarisation. These NPs were further combined with PD-1 checkpoint inhibitors for additive effects [37]. Other studies exploring nano-drug delivery systems for different approaches have been explored further. 

#### Mannose-Targeted PLGA NPs

The expression of mannose receptors on the surface of TAMs can be exploited for targeting using NPs. Niu et al. bioengineered PLGA NPs to actively target TAMs through mannose–mannose recognition [3]. These NPs were loaded with doxorubicin and coated with PEG (DOX-AS-M-PLGA-NPs). The in vivo study displayed high biodistribution of doxorubicin to the tumour sites in mice treated with DOX-AS-M-PLGA-NPs, along with a significant reduction in tumour volumes. The TAMs density was also reduced at the tumour sites, but this was only confined to the TME, as no changes were observed in spleen TAMs. The positive results were initially observed on day 2, and by day 12, TAM levels returned to baseline. However, TAM levels did not rise after repeated functionalised-NP exposure. Insignificant results in tumour volume and TAM reduction were observed in the non-functionalised NPs and the NPs without doxorubicin [3].

### 3.2. Dendritic Cells

Dendritic cells are antigen-presenting cells that are also a part of the innate immune system. Primarily, dendritic cells act as a bridge between the innate and the adaptive immune system when tumours and other problematic cells are identified [38]. The function of dendritic cells is heavily impacted by the tumour type and the environment the cells encounter; some have antitumour effects whilst others are tumour-promoting. The immunosuppressive activity comes about due to the exposure of factors such as VEGF, IL10, prostaglandin E2, and other various cytokines, which prevent cell maturation [38]. Once in the tumorigenic state, they stimulate regulatory T cells, helper T cell 2, and helper T cell 17, aiding in tumour protection [38]. 

Interestingly, once dendritic cells are removed from the highly influential TME, restoration of normal cellular function can be observed. Furthermore, their functionality is related to the stage of cancer, e.g., in the early stages of certain cancers, dendritic cells have demonstrated a tumour-suppressive role but became tumour promoters with disease progression [38]. 

#### Interferon-γ Loaded NPs to Enhance Dendritic Cell Activity

Interferon-γ (IFN-γ) is a pro-inflammatory cytokine that has potent immune-stimulating activity, particularly by stimulating matured DCs, promoting subsequent antitumour activity [36]. A study conducted by Wu et al. investigated cell-derived nanovesicles (NVs) that were co-loaded with doxorubicin, IL-2, and IFN-γ (NV-DOX_IL-2/IFN-γ_) [39]. A greater reduction in tumour volumes occurred when doxorubicin was loaded into the NVs compared with the free drug, signifying enhanced tumour penetration. Treating 4T1 tumour bearing mice with NV-DOX_IL-2/IFN-γ_ led to an 87.9% reduction in tumour volume when compared with free drug and PBS control, indicating a synergistic effect of the combined therapeutic agents [39]. Another study confirmed that combining doxorubicin with anti-PD-1 immune checkpoint inhibitors enhanced dendritic cell activity by causing a seven-fold increase in IFN- γ in comparison to the free drug alone [40]. 

### 3.3. Myeloid-Derived Suppressor Cells

Immature myeloid cells known as myeloid-derived suppressor cells (MDSC) are key players in activating immunosuppressive cells. MDSCs can differentiate into two subtypes: (1) monocytic (M) MDSCs and (2) polymorphonuclear MSDC [41]. It has been well established that MDSCs reduce the efficacy of immune-based therapeutics due to their involvement in innate/adaptive immune inhibition. There is a close correlation between MDSC prevalence and an increased disease burden with a prolonged duration of active disease [5]. 

Attracted to the TME through CCL2, MDSCs exert their effects by secreting numerous cytokines, elevating arginase and prostaglandin E2 levels, and producing reactive oxygen species [9]. The effects extend to suppressing natural killer cells, so targeting this aspect of the TME is hypothesised to provide additive benefits to contemporary treatment [9]. 

#### 3.3.1. Ursolic Acid-Loaded Liposomes’ Actions on MDSC

Ursolic acid (UA) is a naturally occurring pentacyclic triterpenoid found in various fruits, such as apples and basil [42]. It is known to possess potent antifungal and antibacterial effects and recent immune regulatory properties [43]. However, it has very limited clinical application due to its poor solubility. Zhang et al. employed UA in a liposome formulation to modulate the TME [42]. The formulation used in the study was intravenously administered to a murine model of cancer at 10 mg/kg, aiming to improve vascular circulation and enhance accumulation at tumour sites. Gr-1 and CD11b antigens were used as markers of MDSC. Treatment with the liposome formulation resulted in a drastic decrease in biomarker levels in the blood circulation, spleen, and at tumour sites after five doses. This was accompanied by an increase in cytotoxic T cells and, ultimately, greater reductions in tumour volumes across the experimental duration compared with the control [42].

#### 3.3.2. Immune Nanomodulators Targeting MDSCs

Previous research has identified that gemcitabine (GEM) causes necrosis of MDSCs in different cancer microenvironments [44]. A study carried out by Chen et al. used nanocages loaded with GEM in combination with small interfering RNA targeting IDO (siIDO) and program cell death ligand-1 (PD-L1) antibody engineered on the surface of the nanocage [45]. The co-delivery aimed to ameliorate immunosuppression and overall improve TNBC patients’ outcomes. The tri-loaded nanocages (GSZMP) in TNBC-burdened mice led to a reduction in the percentage of MDSCs compared with the control, along with an increase in T cell infiltration. Additionally, tumour volume reduction was the greatest in the GSZMP group with the highest percentage of survival compared with the control, which proved the importance of targeting other factors within the TME [45].

### 3.4. Neutrophils

Neutrophils originate from myeloid progenitor cells and are fundamentally responsible for eliminating invading pathogens through acute inflammation. Once activated through chemokines released by epithelial cells, neutrophils survive for around five days [5,36]. However, proinflammatory factors residing within the microenvironment, such as interferon-γ, increase neutrophils’ lifespan [36]. Inappropriate activation of neutrophils leads to chronic inflammation and tissue damage, ultimately leading to the suppression of critical immune cells [36]. Neutrophils secrete mediators and CCL2 to further recruit more neutrophils and other monocytes and macrophages, respectively. 

Due to the plasticity of neutrophils, their characteristics differ under the influence of neoplastic tumours compared with more evolved cancers. Tumour-associated neutrophil (TAN) phenotypes are classified as N1 (tumour suppressive) and N2 (tumour promoting) [38]. N1 is the subtype present in the early stages of cancer and has a role in activating the immune system through the secretion of cytokines and chemokine, IFN-I, and IL-18. As tumours progress, neutrophils’ phenotype switches to protumorigenic N2 in the presence of regulatory mediators G-CSF or TGF-b [36]. N2′s primary effect is immunosuppression by inhibiting T cells and natural killer cells [46]. The immune escape is facilitated by the secretion of reactive oxidase species, arginase, and peroxidases [36]. Studies show that patients with progressive disease had enriched neutrophils, which were associated with a reduced response to immune check inhibitors (ICI), highlighting the immunosuppressive nature of neutrophils [9]. The outcome of TANs is a poorer prognosis of advanced tumours and a higher tumour reoccurrence. 

There is currently no direct nanoparticle application targeting neutrophils in TNBC; however, there are reports on the indirect mechanisms. Selenium is an immunostimulant that stimulates the function of neutrophils to aid in antitumour activity. Selenium-loaded nanoparticles were used to stimulate the host’s immune response at the tumour sites [47]. The in vivo study showed a tumour volume reduction in the 4T1 mouse model, indicating neutrophil activation [48]. Further studies are needed to elucidate the specific effects of these nanoparticles in targeting neutrophils within the TNBC tumour.

### 3.5. Tumour-Associated Lymphocytes

Cytotoxic CD8+ T cells are powerful components of the adaptive immune system. Fundamentally, they are responsible for destroying invading pathogens and malignant cells within the body [49]. However, within the context of tumours, CD8+ T cell functionality is disrupted by persistent exposure to tumour neoantigens, driving the cell into its dysfunctional state of T-cell exhaustion [50]. This is through the upregulation of cytotoxic T-lymphocyte- associated protein-4 (CTLA-4), a receptor responsible for providing inhibitory signals to control T cell overactivity. Infiltration into the tumour site is also impeded. [51]. 

The TME is a complex interconnected system that involves cellular populations impacting antitumoural activity. Hence, CAFs and TAMs have been found to reduce CD8+ cell infiltration into the tumour site. Both cellular factors create an inhospitable environment for the cytotoxic T cells by creating a dense stroma that is abnormally vascularised [52]. Angiogenic factor VEGF influences the function of CD8+ T cells as it suppresses the recruiting factors (dendritic cells) whilst promoting immunosuppressive cell populations [53,54]. Furthermore, the conditions of the TME impact lymphocyte survival, as it is hypoxic, dense, has abnormal vasculature, and lacks essential nutrients. The hostile factors result in cold tumours, tumours that lack CD8+ cells, allowing tumours to evade immune surveillance. Decreases in CD8+ count correlate to higher TNBC mortality [55].

#### Liposomes Elevating CD8+ Counts

A direct approach aiming to elevate CD8+ count was undertaken by Cheng et al. Cyclic [G (3′,5′) pA(3′,5′) p] (cGAMP) is a known enhancer of the innate and adaptive immune system by targeting intracellular stimulator of IFN genes (STING). cGAMP-encapsulated liposomal NPs (cGAMP-NP) were used in PD-L1-insensitive tumours in a genetically engineered model of basal-like TNBC. The NP application modified various immune factors, including enhancing the infiltration of CD8+ cytotoxic cells into the tumour site. A secondary effect of cGMP-NPs was the switch in polarization from M2 to M1, which also aids in CD8+ infiltration. The net result observed was reduced tumour growth in the treated mice in comparison with the control group among TNBC-burdened mice, indicating tumour suppression. Their findings indicated that cGAMP-NPs can modulate the TME in PD-L1-insensitive TNBC tumours and can produce antitumour memory to prevent secondary tumour development on a single dose, potentially enhancing their responsiveness to immunotherapy [56]. 

## 4. Non-Cellular Component

### 4.1. Extracellular Matrix

The extracellular matrix (ECM) is a non-cellular component of the TME that is composed of basement membranes and an interstitial matrix. The structural network of proteins, glycoproteins, and proteoglycans not only sustains the architecture of the TME, but it also provides nutrients to the surrounding cells [7]. The tumour-induced remodelled ECM is also involved in oncogenic signalling pathways between residing cellular components of the TME. The ability to provide migration pathways for T cells or directly inhibit T cell proliferation acts as a means of either tumour suppression or tumour growth, respectively [37]. The involvement of the ECM results in increased invasion, proliferation, and metastasis of cancer cells. 

The ECM’s structural integrity acts as a physical barrier, preventing drug tissue penetration by increasing density and stiffness. This penetration inhibition effect also extends to antitumour immune cells [57]. The extent of ECM stiffness correlates with a reduction in T cell migration, ultimately affecting tumour behaviour, and is hypothesised to hinder the effectiveness of immunotherapy [58]. An essential factor within the ECM is the overexpression of collagen. Collagen is responsible for increasing the activity of proteoglycans and metalloproteinases, which is driven by an overexpressed enzyme known as Lysyl oxidase (LOX) [37]. The critical function of the enzyme is to covalently cross-link elastin and collagen fibres together, causing a stiffened ECM which further facilitates cell migration and adhesion. Another interactive factor within the ECM is fibronectin, which causes angiogenesis. It exerts its effects by providing survival support to endothelial cells and connects integrins that are responsible for the formation of neovascularisation [7]. Fibronectin is usually expressed in the ECM of non-malignant cellular environments, but the variant extra domain-B fibronectin (EDB-FN) is known to be overexpressed in TNBC [59]. The oncoprotein variant correlates with poor overall survival and is known to be highly abundant in TNBC [59]. 

#### Lipid-Based NPs for LOX Inhibition

Given the contribution of LOX enzymes, De Vita et al. proposed that targeting the enzyme might yield better chemotherapeutic outcomes in TNBC [60]. In this study, lipid-based epirubicin-bearing NPs (Lipo-EPI-LOX) which were bioengineered by LOX blocking antibodies onto their surface were developed. The multifaceted approach aimed to exploit the intrinsic therapeutic activity of targeting LOX in the tumour ECM and selectively accumulate epirubicin within the tumour site with reduced systemic toxicity. Compared with the control (lipid NP alone) and non-functionalised NP, the results showed great potential for Lipo-EPI-LOX. The biodistribution of the anthracycline was localised at the tumour sites, and high internalisation translated to a significant reduction in tumour volumes. Furthermore, the optimised NPs greatly reduced ECM stiffness, signifying LOX inhibition. The outcome of the Lipo-EPI-LOX treatment was prolonged survival and reduced disease progression in TNBC-bearing mice. Thus, the suggested LOX enzyme inhibition serves as a feasible additive approach in treating TNBC [60]. 

Further nanomaterial-based systems that have an impact on different components of the TME and are summarised in Table 1. 

## 5. Conclusions and Future Perspectives

This review describes the impact of the TME as being one of the key contributing factors in the progression and survival of tumour cells. This is especially true for TNBC, as the TME is predominantly responsible for its aggressive nature. Hence, there has been an increased focus on developing targeted therapies aiming at different TME factors. Nanotherapy is a primary example of an emerging drug delivery system explored in cancer therapy via different TME targeting strategies. With the given benefit of controlled drug release rates and high intratumoural accumulation, nanoparticles have been investigated in reducing neoplastic cell populations in TNBC. Many studies have reported positive outcomes when NPs were used in TNBC murine models. Those include significant tumour size reductions and modulation of various TME components. As the microenvironment is interlinked, affecting one cell has been found to have downstream effects on other cellular components. This was seen through either an increase in immunostimulatory factors or decreases in negatively impacting constituents. The application of nanotechnology in this area of oncology is expected to improve TNBC patient outcomes drastically in the clinical setting and lead to a paradigm shift in TNBC therapy. However, clinical translations are yet to be established. A more in-depth preclinical evaluation of different nanotherapeutics is required for clinical translation to occur. 

## Figures and Tables

**Figure 1 pharmaceutics-15-00112-f001:**
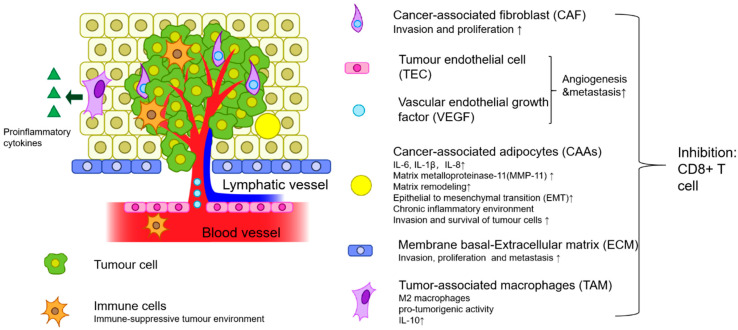
Factors of the TME and their associated impacts on tumour cells. (This figure is reproduced from [6] with modifications to the original source).

**Figure 2 pharmaceutics-15-00112-f002:**
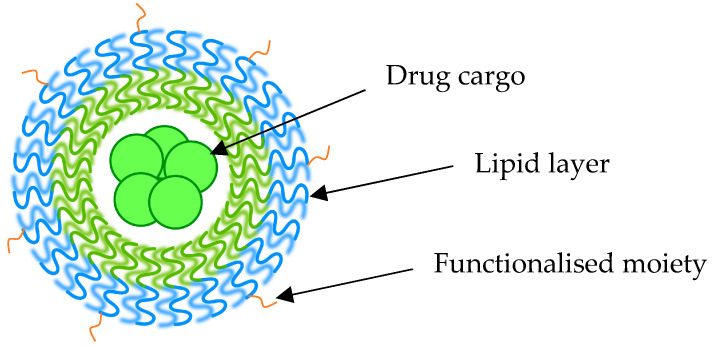
Schematic representation of a functionalised nanoparticle loaded with drug.

**Figure 3 pharmaceutics-15-00112-f003:**
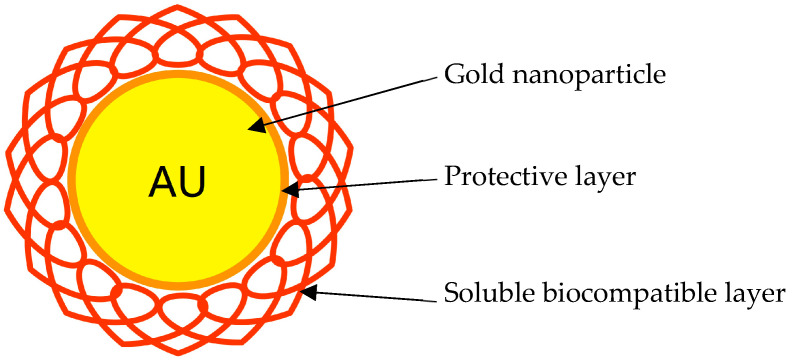
Schematic representation of a gold nanoparticle.

**Figure 4 pharmaceutics-15-00112-f004:**
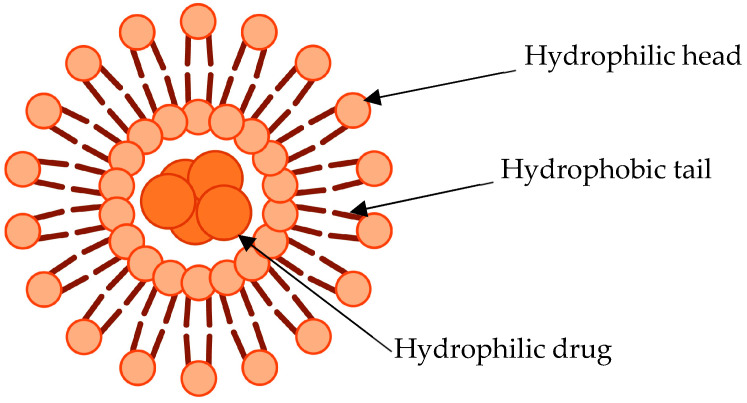
Schematic representation of a basic unilamellar liposome.

**Table 1 pharmaceutics-15-00112-t001:** Different nanoparticles under investigation impacting TME cell populations.

Nanoparticle	Payload	Mechanism of Action	Reference
Cancer-associated fibroblasts			
Spherical nanoparticles engineered with cleavable amphiphilic peptide (CAP)	Paclitaxel	Disruption of stromal barrier and enhanced drug accumulation	[61]
Peptide-derived nanofiber	Losartan	Inhibition of collagen I production and increased therapeutic efficacy	[20]
Ferritin nanocages engineered with Anti-FAP antibody	Photosensitiser	Suppression of chemokine release from CAFs	[62]
Lipid-calcium phosphate (LCP) nanoparticles	Quercetin	Downregulation of Wnt16, CAFs and normalisation of collagen	[63]
LCP nanoparticles	Secreted TNF-related apoptosis-induced ligand (sTRAIL)	Changing CAFs to their quiescent state resulting in tumour growth inhibition	[64]
Mesenchymal stem cells			
Gd-metallofullerenol (Gd@C82(OH)22)	N/A	Reversal of EMT and depletion of cancer stem cells	[65]
Gold nanoparticles coated with poly (ethylene glycol)	Doxorubicin	Inhibition of tumour growth with reductions in stem cells	[66]
Tumour-associated macrophages			
Gold nanorods	Cetuximab	In combination with phototherapy, the enhanced temperature at the tumour site resulted in TAM and tumour death	[67]
Liposomes	17-(allylamino)-17-de- methoxygeldanamycin (17-AAG)	Significant reduction in TAMs with an increase in T cell infiltration and overall reductions in tumour volumes	[68]
Extracellular Matrix			
Programmed site-specific delivery (PSSD) nanosystem	LY3200882 and siPD-L1	Remodelling of the extracellular matrix by reversal of immunosuppression and deeper penetration of nano-based drug delivery systems.	[69]
RGD-PEG-ECO/miR-200c	N/A	Remodelling of the extracellular matrix by targeting miR-200c. Reduction in fibronectin along with suppression of TNBC proliferation.	[70]
Doxil nanoparticle	TGF-β inhibitor tranilast	Reduction of the extracellular matrix density and increased T cell infiltration.	[71]

## Data Availability

Not applicable.

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
