# Peer review of "Potential Nanotechnology-Based Therapeutics to Prevent Cancer Progression through TME Cell-Driven Populations"

_pharmaceutics, 2022, doi:10.3390/pharmaceutics15010112_

Round 1

Reviewer 1 Report

The article entitled “Potential nanotechnology-based therapeutics to prevent cancer progression through TME cell-driven populations” presents interesting findings on the impact of TNBC.

The review is well organized and the figure is suitable for the context and the review. However, I would ask the author to review some spelling mistakes in the whole manuscript as well as additional information in the first row (for what is written in the column) in order to be more clear for the reader.

I suggest acceptance of the paper after the minor revision requested 

Author Response

Authors: Thank you for your comment. We made a thorough language edit to the whole manuscript and added title row for Table 1 as suggested 

Reviewer 2 Report

1.     Targeting the characteristics of tumor extracellular matrix is an important strategy, and is attracting more and more research interest. Therefore, it is advised that this strategy should be included in Table 1. Meanwhile, more publications (For example, Tumor stiffening reversion through collagen crosslinking inhibition improves T cell migration and anti-PD-1 treatment. DOI: https://doi.org/10.7554/eLife.58688) on regulating ECM stiffness should be cited and introduced in Part 4.1.

2.     There is only 1 figure in this manuscript. It is advised that you cite the pictures of the key publications with permission, pictures which demonstrate the overall concept of the publication, such as the graphical abstract. The pictures will make the manuscript easier to read, and more attractive.

3.     In some sections such as 3.5.1, only 1 publication was introduced. It is advised that the manuscript should make the readers understand that a big quantity of publication has been read by you and you concluded something useful after your deep thinking.

4.     It is advised that “4. Conclusion” be changed to “4. Conclusions and Future Perspectives”.

5.     The English and expression should be improved. Suggestions include the following but are not limited to them: Please correct reference 21, which appeared !!! INVALID CITATION !!! [20].. In numerous places, such as Line 52, Line 100, line 201, etc., “compared to” should be changed to “compared with”. Please correct the grammar for “For instance, tumour cells situated away from the vessels get deprived of oxygen and essential nutrients required for uncontrolled proliferation.” (Lines 60-62). “As CSCs overexpress CD44 receptors, they can be used to selectively targeting them.”(Line 208) What do “they” and “them” refer to, respectively? Please clarify. Line 234 Please elucidate the relationship between tumour-associated adipocyte and CCL2.

Author Response

Comments and Suggestions for Authors: Reviewer #2 

1.     Targeting the characteristics of tumor extracellular matrix is an important strategy, and is attracting more and more research interest. Therefore, it is advised that this strategy should be included in Table 1. Meanwhile, more publications (For example, Tumor stiffening reversion through collagen crosslinking inhibition improves T cell migration and anti-PD-1 treatment. DOI: https://doi.org/10.7554/eLife.58688) on regulating ECM stiffness should be cited and introduced in Part 4.1. 

Authors: We agree that the ECM plays a crucial role in the tumour behaviors. We have added additional information in section 4 (Non-cellular component, 4.1 Extracellular matrix), and table 1 of the manuscript, including the above paper.  

2.     There is only 1 figure in this manuscript. It is advised that you cite the pictures of the key publications with permission, pictures which demonstrate the overall concept of the publication, such as the graphical abstract. The pictures will make the manuscript easier to read, and more attractive. 

Authors: More images have been added to the manuscript and a graphical abstract has also been produced. 

3.     In some sections such as 3.5.1, only 1 publication was introduced. It is advised that the manuscript should make the readers understand that a big quantity of publication has been read by you and you concluded something useful after your deep thinking. 

Authors: We had a thorough search strategy and aimed to cover every study that would be relevant to the aims of this review to make it as comprehensive as possible. The reason we only cited one publication in Section 3.5.1   was that this paper was the only citation that looked into this specific area in Triple-negative breast cancer.  

4.     It is advised that “4. Conclusion” be changed to “4. Conclusions and Future Perspectives”. 

Authors: The title has been changed according to the reviewer’s suggestion. 

5.     The English and expression should be improved. Suggestions include the following but are not limited to them: Please correct reference 21, which appeared !!! INVALID CITATION!!! [20].. In numerous places, such as Line 52, Line 100, line 201, etc., “compared to” should be changed to “compared with”. Please correct the grammar for “For instance, tumour cells situated away from the vessels get deprived of oxygen and essential nutrients required for uncontrolled proliferation.” (Lines 60-62). “As CSCs overexpress CD44 receptors, they can be used to selectively targeting them.”(Line 208) What do “they” and “them” refer to, respectively? Please clarify. Line 234 Please elucidate the relationship between tumour-associated adipocyte and CCL2. 

 Authors: The suggested modifications and a thorough edit has been made to the manuscript. Reference 21 is fixed to show valid citations. All editing suggestions have been made. 

Regarding the relationship between adipocytes and the CCL2, this was mentioned in section 2.3, first line, last paragraph. However, to further clarify this point, we modified the sentence to read:  

“TAAs release CC-chemokine ligand 2 (CCL2) through which they cause immunosuppression and further recruit factors into the tumour microenvironment to facilitate tumour survival and propagation.”  

Reviewer 3 Report

·         The authors should add a new paragraph to the introduction to discuss the different types of nano drug delivery systems and their medical application such as improving the bioavailability of drugs also its role for treatment of breast cancer especially resistant breast cancer such as TNBC. Please see these references:

I recommend the authors to add a one or two schematic digram to describe the flow of the review articles. for examples which types of nanosystems are used for each of 

1) mesenchymal originating cells, 2) haematological cells, and 3) non-cellular factors  

Author Response

Comments and Suggestions for Authors: Reviewer #3 

  •          The authors should add a new paragraph to the introduction to discuss the different types of nano drug delivery systems and their medical application such as improving the bioavailability of drugs also its role for treatment of breast cancer especially resistant breast cancer such as TNBC. Please see these references:

I recommend the authors to add a one or two schematic diagram to describe the flow of the review articles. for examples which types of nanosystems are used for each of  

1) mesenchymal originating cells, 2) haematological cells, and 3) non-cellular factors 

  

Authors: Thank you for your comment. The fourth paragraph in the introduction focuses on different benefits of nanotechnology-based drug delivery systems and their medical applications. To address the reviewer’s comment, additional information on various types together with the applications of nanoparticles has been added to the introduction.  

A graphical abstract has also been produced to allow a clearer understanding of the manuscript and describing the flow of the review.  

We had provided Table 1 that summarized different types of NanoSystems used for each of the cellular and non-cellular TME components. However, in response to the reviewers’ suggestions, we added further information to make this table more comprehensive.  

Round 2

Reviewer 2 Report

The manuscript has been improved.